A detailed study of resampling algorithms for cyberattack classification in engineering applications

Mogollón Gutiérrez Óscar oscarmg@unex.es
http://orcid.org/0000-0002-4584-6945 Sancho Núñez José Carlos
http://orcid.org/0000-0002-8717-442X Ávila Mar
http://orcid.org/0000-0002-6367-2694 Caro Andrés
Escuela Politecnica, University of Extremadura , Cáceres, Cáceres , Spain
Cambronero M. Emilia
Electronic publication date: 2024 Apr 15
Publication date: 2024
Volume: 10
Electronic Location ID: e1975
Received 2023 Dec 12; Accepted 2024 Mar 11
Copyright: © 2024 Mogollón Gutiérrez et al.
Copyright year: 2024
Copyright holder: Mogollón Gutiérrez et al.
License: This is an open access article distributed under the terms of the Creative Commons Attribution License, which permits unrestricted use, distribution, reproduction and adaptation in any medium and for any purpose provided that it is properly attributed. For attribution, the original author(s), title, publication source (PeerJ Computer Science) and either DOI or URL of the article must be cited.
License URL: https://creativecommons.org/licenses/by/4.0/

Keywords: Imbalanced learning, Cyber-physical systems, Intrusion detection, Attack classification, UNSW-NB15

Funding: European Union (Next Generation) Detection of Identity Document Forgery using Computer Vision and Artificial Intelligence Techniques C108/23 Instituto Nacional de Ciberseguridad de España (INCIBE) This initiative is carried out within the framework of the funds of the Recovery, Transformation and Resilience Plan, financed by the European Union (Next Generation). The publication is part of the Spanish Strategic Cybersecurity Project “Detection of Identity Document Forgery using Computer Vision and Artificial Intelligence Techniques (C108/23)” funded by Instituto Nacional de Ciberseguridad de España (INCIBE). The funders had no role in study design, data collection and analysis, decision to publish, or preparation of the manuscript.

==============================
The evolution of engineering applications is highly relevant in the context of protecting industrial systems. As industries are increasingly interconnected, the need for robust cybersecurity measures becomes paramount. Engineering informatics not only provides tools for knowledge representation and extraction but also affords a comprehensive spectrum of developing sophisticated cybersecurity solutions. However, safeguarding industrial systems poses a unique challenge due to the inherent heterogeneity of data within these environments. Together with this problem, it’s crucial to acknowledge that datasets that simulate real cyberattacks within these diverse environments exhibit a high imbalance, often skewed towards certain types of traffics. This study proposes a system for addressing class imbalance in cybersecurity. To do this, three oversampling (SMOTE, Borderline1-SMOTE, and ADASYN) and five undersampling (random undersampling, cluster centroids, NearMiss, repeated edited nearest neighbor, and Tomek Links) methods are tested. Particularly, these balancing algorithms are used to generate one-vs-rest binary models and to develop a two-stage classification system. By doing so, this study aims to enhance the efficacy of cybersecurity measures ensuring a more comprehensive understanding and defense against the diverse range of threats encountered in industrial environments. Experimental results demonstrates the effectiveness of proposed system for cyberattack detection and classification among nine widely known cyberattacks.

Introduction

In the past few years, cyber-physical systems (CPS) protection has become increasingly important as new technologies in Industry 4.0 have emerged. Interaction between the “cyber” and “physical” worlds is achieved through the development of industrial systems for many different purposes: food industry (Geng et al., 2022; Avila et al., 2019); autonomous vehicles (Sharma, Sahoo & Puhan, 2021; Hou et al., 2022) agriculture (Drury et al., 2017) or energy consumption forecasting (Khalil et al., 2022; Feng et al., 2023). Artificial intelligence (AI) has recently enabled the development of more complex engineering applications capable of extracting knowledge from the large volumes of data generated by CPS (Radanliev et al., 2020).

Alternatively, artificial intelligence (AI) can be used to ensure a CPS’s cybersecurity by implementing advanced security mechanisms such as intrusion detection systems (IDS) (Monzer, Beydoun & Flaus, 2019). Since CPS manages data vulnerable to malware and cyberattacks, these systems effectively protect CPS from these threats. In this context, cybersecurity plays an important role in protecting assets, services and systems that make up these engineering applications.

To effectively implement an intrusion detection system in a CPS, it is crucial to understand the behavior of the environment. It is possible to extract this behavior from log network connections, traffic flow, network statistics or generated logs, so that it may be determined when the system is being compromised. To accomplish this task, several public datasets are available for simulating a CPS scenario over a controlled time. Both normal traffic and malicious traffic are collected during this period.

Network traffic datasets are usually dirty and unstructured, and, frequently, there is an imbalance in the number of samples belonging to each class that needs to be addressed (Japkowicz & Stephen, 2002). Thus, in a binary classification, having a different number of samples in each class is usual. Problems arise from the imbalance between classes when the difference between the cardinality of the datasets is vast. Higher detection rates may be achieved with clean and balanced datasets. For this reason, problems derived from imbalanced learning have been discussed and studied for years (Branco, Torgo & Ribeiro, 2016; Moniz, Branco & Torgo, 2017; Sancho et al., 2020).

Consequently, analyzing and accurately classifying network traffic into various categories is a challenge in cybersecurity. Numerous efforts have been devoted to this issue (Al-Garadi et al., 2020; Kilincer, Ertam & Sengur, 2021). The usual goal is to identify attacks by analyzing traffic to prevent and mitigate their terrible consequences. The challenge is even more significant because, in real environments, the different classes will be unbalanced: the vast majority of real traffic corresponds to legitimate traffic. The classes corresponding to the different attack types are usually in the minority.

So far, previous studies based on network traffic analysis have inclined to develop classification or prediction algorithms (Al-Garadi et al., 2020; Kilincer, Ertam & Sengur, 2021). However, the impact of resampling techniques on the performance of their proposal needs to be studied more in-depth. In these works, increased importance is devoted to classification techniques rather than resampling strategies. On the other hand, scientific contributions addressing the class imbalance problem usually propose slight variations to well-known resampling algorithms or develop new resampling methods suitable for a single set of data. They limit themselves to testing a single learning system (Rani & Gagandeep, 2022; Bagui & Li, 2021; Khan et al., 2019; Zhang, 2004). In the case of testing several classification algorithms, only results with a single balancing strategy are shown (Aziz & Ahmad, 2021).

This article aims to provide an overview of the state of the art of imbalanced learning oriented to network traffic among industrial systems. A complete study of oversampling and undersampling techniques is presented to reduce the problem of imbalanced datasets in this field. This study will lay the foundations for the proposed classification model. Specifically, the three most common oversampling techniques (Synthetic Minority Oversampling Technique (SMOTE), SMOTE-Borderline1 and Adaptive Synthetic) and five widely used undersampling techniques (random undersampling, cluster centroids, NearMiss, repeated edited nearest neighbor, and Tomek Links) are studied.

Regarding the classification model, this proposal consists of a two-phase ensemble classification model. This scheme has been adopted due to the irregular distribution of samples per class. Following the proposed framework in Mogollón-Gutiérrez et al. (2023), attack detection is performed before attack classification. This contribution demonstrates the effectiveness of this solution in a complex network traffic environment.

Thus, the novelty of this study lies in balancing a multi-class dataset to evaluate the performance of several classification models in CPS. To the best of our knowledge no previous studies have focused their efforts on comprehensively studying the impact of sample balancing on cybersecurity-oriented datasets for the protection of CPS. A novel approach is described to mitigate the imbalanced learning problem of network traffic analysis and classification. The article illustrates how to deal with an actual critical application of an imbalanced domain. Indeed, anomaly detection solutions are in high demand in the industry, especially for intrusion detection, privacy, and security.

The main contributions of this research can be summarized as follows: 1) A cybersecurity model for anomaly detection and classification in CPS is proposed. The simulation of traffic over a CPS is conducted using generic network traffic consisting of nine different cyberattacks and benign traffic. To evaluate the effectiveness of the proposed model, performance will be compared with some state-of-art works.

2) A novel dataset generation for imbalanced classification problems. This process is based on eight well-known resampling algorithms (three for oversampling and five for undersampling) and generates one-vs-rest datasets for each traffic category.

3) Design of a two-stage multi-class classification model. In the first approach, binary models are generated by selecting one oversampling algorithm and one undersampling algorithm, resulting in a total of 16 models that are generated from 3 × 5 resampling combinations without resampling the dataset. Next, a second approach is developed, selecting the resampling technique that performs best for each category. This latter approach is considered the proposed classification model.

The remaining parts of this article are organized as follows. “Related Work” briefly surveys cybersecurity res and existing contributions on imbalanced problems. The dataset, resampling strategies and classification algorithms used in experiments are presented in “Method”. “Experimental Design” describes binary dataset generation according to different resampling techniques and the two-stage classification model. “Results and Discussion” discusses the results obtained by our approach and a performance evaluation. Finally, “Conclusions” draws the conclusions and some future research directions.

Related work

Network traffic gathering is a fundamental task for CPS protection elements, such as intrusion detection systems. A collection of data with enough information can help build intelligent systems capable of distinguishing different types of traffic and, consequently, take measures for the correct operation of the protected asset.

To help in this field, multiple researchers have generated datasets that keep different states of a network. The monitored traffic corresponds to legitimate or attack behavior; in the latter case, the type of attack that has occurred is indicated. One of the datasets with the most significant impact on the scientific community is KDD99 (Defense Advanced Research Projects Agency (DARPA), 1999). This dataset collects four types of cyberattacks together with Normal behavior. One of the greatest challenges in this dataset is duplicate behavior due to the data collection method based on synthetic traffic generation.

Tavallae addressed this problem in Tavallaee et al. (2009) with the NSL-KDD dataset. This research attempted to solve this problem by applying statistical analysis, allowing redundant data removal and identification of the most difficult-to-learn categories. Both tasks result in an improved version of the previous dataset.

Among recent datasets with the highest impact are UNSW-NB15 (Moustafa & Slay, 2015) and CIC-IDS2018 (Sharafaldin, Lashkari & Ghorbani, 2018). In both datasets, the traffic collection is carried out by simulating several attack scenarios. Specifically, collected traffic in UNSW-NB15 is distributed among 10 categories, one for normal traffic and the rest corresponding to attacks that may compromise the information system security. In the case of CIC-IDS2018, five different attacks were collected.

A common problem with the reviewed datasets is the distribution of records. Most of the samples belong to legitimate behavior. In the case of KDD99, the number of samples from the normal class is 812,814 in the training set vs 52 samples for the minority class. Considering the minority class as the category with fewer samples results in an imbalance ratio of 15,655. The metric of the imbalance ratio serves as a crucial indicator of the degree of imbalance within a dataset. A high imbalance ratio signifies a substantial disproportion between the minority and majority classes, highlighting potential difficulties in model learning where algorithms might become biased towards the majority class. This problem is not solved in the improved version, NSL-KDD, where the category corresponding to legitimate traffic consists of 67,342 vs 52 samples for the minority class, resulting in an imbalance ratio of 1,295. Regarding UNSW-NB15, an imbalance ratio of 430 is obtained, having 56,000 samples for legitimate behavior and 130 for the minor class attack. The highest imbalance ratio is obtained in CIC-IDS2018, where the benign sample size is above 6 million and the minority class does not reach 1,000 samples.

To address this issue, resampling algorithms can generate new synthetic samples or remove data that meet some criteria. According to Ali, Shamsuddin & Ralescu (2015), these strategies can be classified into two types. The techniques of the first group, data-level, consist of balancing the number of samples of each category using preprocessing techniques such as Synthetic Minority Oversampling Technique (SMOTE) (Chawla et al., 2002) or cluster-based sampling (Nwe & Lynn, 2020; Lin et al., 2017; Hong et al., 2009). The second group, algorithm-level, is composed of a set of algorithms capable of learning the distribution of an unbalanced dataset, such as one-class learning (Devi, Biswas & Purkayastha, 2019), improved algorithm (Wang & Sun, 2021), cost-sensitive learning (Yu et al., 2018); ensemble (Hayashi & Fujita, 2021) and hybrid techniques (Ayyagari, 2020). Some relevant contributions that use resampling techniques: SMOTE is used in Li, Abdel-Aty & Yuan (2020), Ahmed, Hameed & Bawany (2022) to train a real-time traffic accident detection model. In the work presented by Khan et al. (2018), a cost-sensitive deep neural network is constructed to improve learning on imbalanced image datasets. An ensemble approach is presented in Haixiang et al. (2016) that combines feature selection through binary particle swarm optimization for feature selection and boosting k-nearest neighbor classifier.

SMOTE is one of the main oversampling techniques, resulting in a starting point for developing new techniques that may improve performance depending on the nature of the data. For example, a new algorithm whose novelty lies in reducing noise using support vector machine (SVM) and modifying the equation for generating new samples is proposed in Liang et al. (2020). This proposal is an improvement over SMOTE on a medical dataset. Another novel proposal is presented in Maldonado et al. (2022). In their work, the weighted Minkowski distance is calculated to determine the neighborhood of minority samples and, thus, prioritize the most relevant features of an unbalanced dataset.

Undersampling algorithms are organized into two subcategories according to Triguero et al. (2012). On the one hand, there are methods capable of reducing the number of samples by generating new samples representative of the majority class (prototype generation). The work carried out by Ren & Yang (2019) propose a clustering-based algorithm to generate new samples for both majority and minority classes. On the other hand, some methods select a subset of samples based on a heuristic (prototype selection). In Xouveroudis et al. (2021), sample selection based on condensed nearest neighbors algorithm is proposed.

In the literature, several studies show that resampling unbalanced datasets generates more robust classification systems. In Lin et al. (2021), a classifier based on Random Forest ensemble is proposed for intrusion detection to a renewable energy system. This research applies SMOTE-based oversampling techniques to resample the classes and minimize the classification error. Another ensemble solution based on LightGBM is addressed in Liu, Gao & Hu (2021), where new synthetic samples are generated with adaptive synthetic sampling (ADASYN). This approach represents an improvement of 1.91% concerning the construction of the same model with the unbalanced ensemble.

In the field of deep learning, autoencoders (AEs) have emerged as a reliable tool in anomaly detection tasks, especially within the realm of network security. These models are designed to learn efficient representations of data by compressing input data into a lower-dimensional space (encoding) and then reconstructing the input data from this representation (decoding). For instance, in Fu et al. (2022) deep learning model integrating a stacked autoencoder for dimensionality reduction, enhancing information fusion and improving detection accuracy in imbalanced datasets. A novel five-layer AE model, focusing on optimal data preprocessing and reconstruction error function, leading to superior detection accuracy and model performance, is presented in Gu et al. (2023). A deep denoising autoencoder within a multi-module intrusion detection system to extract deep features and cope with data imbalance and high dimensionality in developed in Xu et al. (2021). Proposed multi-module integrated system is presented in Cui et al. (2022), utilizing a stacked autoencoder for feature extraction, which, along with advanced processing and classification modules, achieves significant improvements in detecting unknown attacks and reducing false alarms.

In cyberattack classification, Agarwal et al. (2021) evaluates machine learning algorithms focusing on accuracy and processing time reduction. Rani & Gagandeep (2022) proposes an intrusion detection system based on a cost-sensitive neural approach, obtaining an improvement of about 4% over baseline. In IoT, deep learning methods can more effectively improve the accuracy of IoT attack detection under unbalanced samples, as shown in Zhang & Liu (2022). The performance of an intrusion detection system is evaluated in Bagui & Li (2021) by applying random undersampling, random oversampling, SMOTE and ADASYN methods. These resampling methods are applied to train an artificial neural network. In Divekar et al. (2018) SMOTE and Random Undersampling are applied for UNSW-NB15 and NSL-KDD binary classification. Two-class classification is also presented in Aziz & Ahmad (2021) with an unsupervised undersampling approach using k-means. Legitimate traffic is oversampled in Khan et al. (2019), and a two-step deep learning solution is presented. However, this study does not show any results without oversampling the dataset. In Zhang et al. (2020) a preprocessing technology that combines SMOTE and clustering undersampling is developed together with a convolutional neural network. One main issue with their contribution is that they do not provide classification metrics that match the imbalanced problem.

In summary, some studies apply resampling techniques that demonstrate their effectiveness in numerous fields. For this reason, protecting CPS against cyberattacks is an ideal field of application to study the impact of these techniques. In the works reviewed, the authors limit themselves to testing a resampling strategy with one or several classification algorithms or several balancing techniques with a single machine or deep learning approach. In addition, some of these studies focus on attack detection rather than including results related to multi-class classification. For these reasons, the present work focuses on performing a comprehensive analysis of various oversampling and undersampling techniques on an unbalanced network traffic dataset. To demonstrate their effectiveness, multi-class and binary classification is evaluated using a two-stage learning methodology. This approach requires the combination of multiple individual binary classifiers whose predictions are combined to determine the final category of a sample. Furthermore, evaluation metrics for imbalance problems are considered when discussing the results.

Method

This article evaluates the influence of resampling in the development of a system to identify and classify the network traffic based on ten different traffic categories. To contextualize this research, a brief description of the dataset and its features is provided. Next, the resampling methods employed are explained, along with a graphical representation of their impact on the 2D feature space of a generic dataset. Finally, the classification algorithms are briefly described together with the evaluation metrics used to assess the classification capability according to the resampling techniques.

Dataset

The UNSW-NB15 dataset was created by the Cyber Range Lab at UNSW Canberra to simulate a heterogeneous environment of legitimate traffic and real attack traffic (Moustafa & Slay, 2015). The original dataset consists of 2,540,044 tuples and exhibits a high-class imbalance, as only the distribution of Normal type tuples accounts for more than 87% of the total samples. However, as previously indicated, the authors have published a row and feature reduced dataset (Moustafa, 2015). This latter set is already divided into test and trial sets, where an imbalance between the number of samples from different traffic classes is also maintained. The experiments in this research have been performed with the reduced dataset. Table 1 shows the 44 features and their type, of which 40 are numerical and four categorical (proto, service, state and attack_cat). Distribution of classes for the training and test sets is specified in Moustafa & Slay (2015).

Table 1 Dataset features and their types.

Feature	Information	Type	Feature	Information	Type	
proto	Flow	Category	swin	Content	Integer	
dur	Basic	Decimal	dwin	Content	Integer	
service	Basic	Category	stcpb	Content	Integer	
state	Basic	Category	dtcpb	Content	Integer	
spkts	Basic	Integer	smean	Content	Integer	
dpkts	Basic	Integer	dmean	Content	Integer	
sbytes	Basic	Integer	trans_depth	Content	Integer	
dbytes	Basic	Integer	response_body_len	Content	Integer	
rate	Basic	Decimal	ct_srv_src	Generated	Integer	
sttl	Basic	Integer	ct_state_ttl	Generated	Integer	
dttl	Basic	Integer	ct_dst_ltm	Generated	Integer	
sload	Basic	Decimal	ct_src_dport_ltm	Generated	Integer	
dload	Basic	Decimal	ct_dst_sport_ltm	Generated	Integer	
sloss	Basic	Integer	ct_dst_src_ltm	Generated	Integer	
dloss	Basic	Integer	is_ftp_login	Generated	Integer	
sinpkt	Time	Decimal	ct_ftp_cmd	Generated	Integer	
dinpkt	Time	Decimal	ct_flw_http_mthd	Generated	Integer	
sjit	Time	Decimal	ct_src_ltm	Generated	Integer	
djit	Time	Decimal	ct_srv_dst	Generated	Integer	
tcprtt	Time	Decimal	is_sm_ips_ports	Generated	Integer	
synack	Time	Decimal	attack_cat	Label	label	
ackdat	Time	Decimal	label	Label	Integer	

The rationale behind choosing this dataset is that in comparison to the previously mentioned datasets, not only does UNSW-NB15 have the most realistic network traffic, but it also contains the highest variety of cyberattacks compared to a more recent IDS dataset, CIC-IDS2018 (Sharafaldin, Lashkari & Ghorbani, 2018), which only captures five different types of attacks.

Researchers who created the UNSW-NB15 group dataset feature in five groups depending on the information kept. Flow features. They represent the flow features extracted from the UNSW-NB15 dataset, including source and destination IP addresses and port numbers. Although these features are fundamental, they are not included in the reduced dataset to focus on more analytically valuable features. This table sets the groundwork for understanding the basic structure of network traffic data being analyzed.

Basic features. They detail the basic features of network transactions captured in the UNSW-NB15 dataset. These features include the state of the transaction, duration, bytes transmitted between source and destination, packet counts, and service type. This diverse set of features is crucial for initial analysis and detection models, offering insights into the general behavior of network traffic and potential anomalies.

Content features. They focus on content features, which include TCP window sizes, sequence numbers, packet sizes, transaction depths, and response body lengths. These features dive deeper into the network traffic content, providing a richer context for anomaly detection by examining the specifics of the transmitted data and the characteristics of individual packets.

Time features. They introduce time-based features, such as source and destination jitter, packet inter-arrival times, and TCP connection setup times. Time features are vital for identifying anomalies that manifest through unusual time patterns, such as delays or rapid sequences of activities, which might indicate malicious behavior or network issues.

Generated features. They list generated features, which are derived from the analysis of network traffic, including binary flags for specific conditions (e.g., FTP logins), counters for connection patterns, and transaction statuses. These features represent higher-level abstractions and summaries of the traffic, focusing on behavioral patterns that help distinguish between normal and anomalous activities.

Resampling techniques

A synthetically generated dataset was used to visualize the effect of each of the methods applied in the research. It can be visualized in Fig. 1. It consists of a three-class dataset with a total o 479 samples. One of them being the majority class with 379 samples. The remaining two are composed of 50 samples. To understand each resampling technique, Fig. 2 shows the original dataset and the effect of the resampling algorithms on it. Depending on the applied approach, each class’s final number of samples is highlighted in red and green when oversampling or undersampling, respectively.

Figure 1 Original synthetic dataset for resampling visualization purpose.

Figure 2 Resampling comparison.

From a cybersecurity perspective, resampling actual traffic results in synthetic traffic, which is helpful for two main reasons: on the one hand, it allows dealing with imbalance by generating traffic similar to that observed in real networks. On the other hand, it enables CPS attack simulations that exploit previously unknown hardware, firmware, or software vulnerabilities, commonly known as 0-day attacks. CPS systems containing many heterogeneous devices are more likely to suffer these attacks. Oversampling. Oversampling methods seek to generate more information from minority classes in a balanced dataset to increase the number of samples and facilitate identifying decision boundaries between classes. – Synthetic Minority Oversampling Technique (SMOTE). This method is one of the most popular when working with unbalanced datasets. This algorithm is based on generating synthetic samples of the minority classes from the nearest neighbor samples. Two factors are involved when generating synthetic samples using SMOTE: the number of neighbors to be considered during generation and the sampling factor that will govern the number of samples generated in the minority class. The detailed procedure is described in Chawla et al. (2002) and is summarized as follows: for each sample in the minority class, its k-nearest neighbors are computed and, depending on the sampling factor, a subset of neighbors is randomly selected. For each selected neighbor, a synthetic sample is generated from the difference of feature vectors between the sample to be treated and its neighbor.

– SMOTE-Borderline1 (BLINE-1). In classification problems, the calculation of the decision boundaries of a class consists of estimating a function that separates the samples belonging to that class from the rest. Given the complexity and quantity of available data, finding the optimal decision boundaries is a complex task that requires a high computational time. In addition, the existence of unbalanced datasets or borderline samples must be added. The SMOTE-Borderline1 method is a variation of the original SMOTE method aimed at identifying class boundaries by generating synthetic samples. The algorithm proposed by Han, Wang & Mao, 2005 and the basic idea consists of the generation of synthetic samples of the minority class that are surrounded to a greater extent by samples of the majority class among their k neighbors.

– Adaptive synthetic (ADASYN). Another alternative is proposed in He et al. (2008) when analyzing highly imbalanced datasets. This sampling technique increases the number of samples in the minority class by shifting the decision boundary toward those samples that are more difficult to learn. In this way, the ability to classify minority samples is improved. The algorithm works as follows: the first step determines the number of total synthetic samples to be generated. Then, the calculated number of synthetic samples for each of the minority class samples is generated. This number is obtained by the number of samples belonging to the majority class among its k-nearest neighbors. Finally, for each minority sample, a neighbor belonging to the minority class is chosen and a synthetic sample will be generated from the difference between their feature vectors.

Undersampling. These methods aim to reduce the number of samples in the training set while maintaining the classification accuracy of the final model. – Random undersampling (RU). The random undersampling method is the most basic technique for balancing a dataset. Specifically, this method randomly removes samples from the majority class until the classes are balanced. It is important to note that this method only works to decrease the number of samples in the majority classes and does not introduce any data variety after removing samples.

– Cluster centroids (CC). This algorithm is based on generating a new set of samples from the class to be resampled. Unlike the other undersampling methods, no selection criterion is applied to decide which samples become part of the final training set. Nevertheless, the undersampling will be performed by generating fewer new samples. The cluster centroids technique uses the unsupervised KMeans algorithm to estimate the clusters of the original set and, once identified, replace the original samples with the centroids of the class to which they belong.

– NearMiss (NM). The literature describes three heuristics for sample selection based on Nearest Neighbors algorithm. These are presented in Mani & Zhang (2003). In NearMiss-1 (NM1), a subset of samples from the majority class is selected such that the average distance between each majority sample and the nearest minority neighbors is the smallest. In NearMiss-2 (NM2), a subset of samples from the majority class is selected such that the average distance between the processed sample and the farthest minority neighbors is the smallest. Finally, the NearMiss-3 (NM3) heuristic consists of two steps: the first step is to search for a specified number of majority class neighbors for each minority class sample. In the second step, the selected majority of samples are the farthest away from their neighboring samples. Both NearMiss-1 and NearMiss-2 can be influenced by noise, but NearMiss-3 maintains close majority samples due to the first-stage selection, where it is ensured that several samples of the minority class surround each majority sample. For this reason, the experiments of this research employ the latter heuristic (NM-3).

– Repeated edited nearest neighbour (RENN). This undersampling method is based on the repeated application of the edited nearest neighbours (Wilson, 1972) algorithm. It can be summed up as follows: for each majority sample, its K nearest neighbors are calculated, and a selection criterion will be applied to determine whether the sample is kept. There are two selection criteria: the first one considers that the majority sample should be removed in case the majority of its k-nearest neighbors do not belong to the same class; the second, more restrictive, implies that all k-neighbors must belong to the class of the processed sample; otherwise, it will be removed. In the case of RENN, the undersampling strategy applied in this research, the repetition of the algorithm implies further sample elimination.

– Tomek Links (TKL). This technique is based on the concept of Tomek’s link. The explanation of this concept and the detailed implementation of the algorithm are revised in Tomek (1976). A pair of samples meeting two criteria in a dataset is called a Tomek link: (i) the samples belong to different classes and (ii) they are mutually nearest neighbors. Applying undersampling based on Tomek’s links may be performed by eliminating both samples or the one belonging to the majority class. For experiments carried out in this contribution, a second criterion is applied.

Table 2 shows the configuration used for each resampling method. Resampling is performed using Python 3.8 with Pandas, Numpy and Imbalanced-learn (Lemaître, Nogueira & Aridas, 2017) libraries. To decide the resampling strategy, the approach described in “Experimental Design” has been applied when the algorithm is capable of controlled resampling. Otherwise, the auto value of the sampling_strategy parameter was chosen. Following algorithms do not allow to set a custom resampling factor: ADASYN, repeated edited nearest neighbors and Tomek Links.

Table 2 Parameters of resampling techniques.

Technique	Parameters	
SMOTE	Sampling_strategy = custom resampling k_neighbors = 5 random_state = 42	
BorderlineSMOTE	Sampling_strategy = custom resampling k_neighbors = 5 m_neighbors= 10 kind = ‘borderline-1’ random_state = 42	
ADASYN	Sampling_strategy = auto n_neighbors = 5 random_state = 42	
Random undersampling	Sampling_strategy = custom resampling replacement = False random_state = 42	
Cluster centroids	Sampling_strategy = custom resampling voting = auto random_state = 42	
NearMiss	Sampling_strategy = custom resampling version = 3 n_neighbors = 3 n_neighbors_3 = 3	
Repeated edited nearest neighbors	Sampling_strategy = auto n_neighbors = 2 max_iter = 100 kind_sel = all	
Tomek links	Sampling_strategy = auto	

Classification algorithms

For the evaluation of resampling algorithms, classical machine learning models have been trained. Each algorithm was selected based on its distinct characteristics and proven effectiveness in handling various aspects of cyberattack detection. SVM (Hearst et al., 1998) is renowned for its robustness in high-dimensional spaces, making it ideal for complex attack patterns (Ahmad et al., 2018). Decision trees (Quinlan, 1986) provide interpretability, which is crucial for understanding the decision-making process behind classifying types of cyberattacks (Ferrag et al., 2020). KNN (Guo et al., 2003) was chosen for its simplicity and efficacy in instances where attacks form identifiable clusters (Alharbi et al., 2021). Lastly, Naive Bayes (Zhang, 2004) offers fast prediction capabilities, especially useful for large datasets typical in cybersecurity (Gu & Lu, 2021). Later, these models are combined to build a more robust solution, which will be described in-depth in “Experimental Design”.

Evaluation metrics

Our study evaluates intrusion detection models’ performance using several metrics. Among the metrics used are accuracy, precision, recall/detection rate (DR), F1-score, and geometric mean (G-mean), which are calculated based on true positives (TP), true negatives (TN), false positives (FP), and false negatives (FN). Each metric is described in Mogollón-Gutiérrez et al. (2023) together with its calculation using confusion matrix.

Experimental design

This section provided an in-depth explanation of the experimental design to define CPS’s network traffic detection and classification model. The following subsections describe the whole process, from data preprocessing to the implementation of the final solution, detailing in-depth the generation of specialized binary models through balancing techniques.

Figure 3 illustrates the phases of the experimental design. Starting from the UNSW-NB15 dataset, the classification algorithms and resampling techniques studied, the main phases of the research are the generation of binary classification models and the construction of a two-phase classification model from the trained models. The generation of binary classification models is detailed in “Generation of Classification Models”. It roughly consists of generating a binary model for each algorithm and considering a pair of resampling techniques. This results in many binary models (3 oversampling × 5 undersampling × 4 algorithms × 10 categories), which, together with the models generated from raw data (4 algorithms × 10 categories), results in a total of 640 models. The two-phase model building, described in “Two-Stage Classification Model”, combines the models generated in the previous phase to build a network traffic classifier capable of distinguishing between legitimate traffic and nine cyberattacks.

Figure 3 Experimental design.

Dataset preproccesing

As usual, the data must be prepared prior to applying classification algorithms. This task has been meticulously performed using Python 3.8, alongside Pandas and Scikit-Learn for efficient data handling and preprocessing (Pedregosa et al., 2011). To ensure the integrity and quality of our dataset, we implemented thorough validation and cleaning steps, including outlier detection and handling missing values. Numerical features were normalized to a standard score, achieving a Gaussian distribution with mean zero and unit variance, thereby enhancing model comparability and performance. Categorical features underwent ordinal encoding, carefully chosen to maintain the inherent structure and significance of the data, thereby avoiding the introduction of ordinal assumptions where inappropriate. We confirmed the absence of redundant samples, as highlighted in Moustafa & Slay (2016), further ensuring the robustness of our dataset. To prevent data leakage, a rigorous approach was adopted where preprocessing steps were applied within each fold of the cross-validation process, ensuring that our model evaluation remains unbiased and reflective of true predictive performance. Figure 4 graphically details our preprocessing workflow, underlining our commitment to data integrity and the reproducibility of our research findings.

Figure 4 Feature preprocessing in UNSW-NB15.

Generation of classification models

Due to the nature of the UNSW-NB15 dataset, there is a clear imbalance with a majority presence of legitimate behavior. This fact may have a negative impact on the performance of a CPS traffic classifier in a production environment. Our proposal addresses this issue by using a binary model system that allows for the classification of network traffic among the ten classes considered. For this purpose, ten binary classifications are built to distinguish the traffic of one category from the rest, following a one-vs-rest approach. These models are generated in a balanced way considering previous resampling algorithms. Thus, the goal of this step is the generation of 10 binary models for each classification algorithm. This is repeated for each pair of resampling techniques (3 × 5 resampling combinations) and without resampling the dataset, resulting in 16 grouped experiments. In the following step, these models will be combined to generate two-step ensemble models.

To better understand the model generation process, Fig. 5 shows a step-by-step diagram of the tasks that must be performed to generate a specialized binary model associated with a traffic category.

Figure 5 Binary specialized model generation.

Starting from the UNSW-NB15 dataset, generating a specialized binary model is as follows: the first step is to create a binary dataset composed of samples of the target category and the rest. The number of samples of the target category in the new dataset is the number of samples in the original dataset. The number of samples in the target category is required ( n_samples) to determine the number of samples for the rest of the categories. The value obtained is divided by the number of categories minus one ( n_samples_c). Since UNSW-NB15 consists of ten categories, it is always divided by nine. n_samples_c will serve as a threshold to decide the resampling type for each category. In case n_samples>n_samples_c means there are more samples than expected, the undersampling method will be applied. When needing more samples, n_samples<n_samples_c, the preprocessed category is oversampled. Once all categories are resampled, together with the samples of the initially selected category, a new resampled binary dataset is generated. As can be appreciated, one oversampling technique and one undersampling technique are used.

The second step consists of the generation of the binary model. For the four classification algorithms reviewed above, a hyperparameter tuning, together with preprocessing, is performed to generate, for each algorithm, the model that best fits the dataset. This optimization uses Grid Search and five-fold cross validation methodology, taking the macro-F1 metric as an evaluation criterion during the hyperparameter fitting process. The reason for considering macro-F1 is that the new dataset has two main classes; therefore, both classes are considered equally important. Finally, after having trained four models, we proceed to select the best model based on the evaluation metrics explained in “Evaluation Metrics”. At this point, a specialized binary model for one category is generated. This process must be repeated for each category in dataset and each pair of resampling algorithms.

To sum up, 640 experiments are required. Since three oversampling techniques and five undersampling techniques have been studied, a total of fifteen different combinations have been tested. In addition, the possibility of not using any resampling technique is also considered, resulting in sixteen possible combinations of experiments. These combinations train the classifiers for each of the 10 types of network traffic considered. Furthermore, the process is repeated for each of the four classification algorithms, generating 640 classification models. Among all of them, and according to the performance metrics considered, the final system includes the best model for each network traffic class: the final system includes ten classification models.

The experiments generate balanced training datasets by applying oversampling and undersampling techniques. Some resampling methods can be configured by including the cardinality of the resulting datasets. That is, the number of samples desired after applying the resampling techniques can be configured in some resampling algorithms but not others. Then, the class distribution of the resulting datasets is computed as follows: For the resampling methods where the final cardinality can be defined, the datasets used to generate the binary models are generated in a balanced way. Thus, for each model, half of the samples belong to one category, and the other half is composed of traffic samples from the other nine categories. This second half is configured to contain the same number of samples from each of the remaining nine categories, where all categories have the same weight (same cardinality) in the second set of samples. For example, according to dataset distribution, the Backdoor category consists of 1,746 samples, so the resulting training set for the generation of the binary model consists of 3,492 tuples: half of the samples of the “backdoor” type (1,746), and the other half (1,746) of “not backdoor” type. This second half is built with the same cardinality (194) for each of the nine categories. To reach the 194 samples, oversampling or undersampling tasks must be performed on the category processed. The result of this process is shown in Fig. 6 for the “backdoor” category.

For the resampling algorithms where the resulting number of samples cannot be controlled, the decision to oversample or undersample is based on the same idea. In this case, the number of samples in each category is calculated, and then oversampling is applied to categories with fewer samples and undersampling to those with more samples than the number obtained.

Figure 6 Distribution of categories for generation of specilized binary model for analysis.

Algorithm proposed in Mogollón-Gutiérrez et al. (2023) shows the process of generating training sets for all dataset categories. In Fig. 5, the activities that are part of the highlighted section should be repeated for each category of the original dataset. This results in a specific dataset for each traffic category generated from an oversampling and an undersampling technique.

Once the specialized datasets are generated, hyperparameter tuning for binary models is carried out using GridSearch and five-fold cross-validation using Python 3.8 and Pandas, Numpy and Scikit-Learn (Pedregosa et al., 2011) libraries. This implies that, for each algorithm, all possible combinations of parameters will be tested over five partitions on the original training set, thus, estimating the optimal values for each parameter. Table 3 shows the hyperparameter search space for each of the applied algorithms.

Table 3 Grid of tunable hyperparameters for classification algorithms.

Algorithms	Hyperparameters	
Naïve bayes	Var_smoothing = 1×10−20…5	
KNN	n_neighbors = 1…30 weights = uniform,distance algorithm = auto,ball_tree,kd_tree	
SVM	Kernel = rbf,linear C = 0.1,1,10,100 gamma = 1×10−5…1	
DT	Criterion = gini,entropy max_depth = 1…35 min_samples_split = 2…5 min_samples_leaf = 1…5	

Two-stage classification model

With 640 models already generated, the next step is to group binary models by the pair of resampling techniques applied for its generation. Thus, 40 models are generated for each one. These binary models, in turn, are divided into 10 categories. In addition, for each category, four classification algorithms are trained.

In order to evaluate the performance of each resampling method, a criterion must be established to select the binary model (out of four) that performs the best in each category. This choice is based on the evaluation metrics obtained individually, specifically macro-F1. As previously explained, this metric considers both classes equally and does not assume the number of samples that make up each class. The selected model becomes part of the two-stage model for a pair of resampling algorithms. In total, 15 approaches will have to be evaluated and compared with the baseline approach. This baseline performance is obtained through the combination of non-resampled binary models.

Following the same idea, an improved solution is proposed. It is based on selecting the best binary model for each category among 16 approaches. Thus, each model is selected with the resampling techniques that yield the best results. This new approach aims to compare its performance with the baseline solution and find the resampling strategies that best fit each kind of traffic.

The construction of the system for both approaches is the same, and it is represented graphically in Fig. 7. It is based on a two-step classification model organized into two phases, as described in Mogollón-Gutiérrez et al. (2023). In this framework, the first classification level aims to detect suspicious traffic. Then, the second classification level is intended to classify the cyberattack among labeled categories.

Figure 7 Proposed two-step scheme for multi-class classification.

Results and discussion

This section presents the results of the resampling experiments applied on UNSW-NB15 after generating 640 binary models. The first set of results shows the performance of each binary model according to the algorithm and the pair of resampling techniques applied. These models are then combined to build the two-stage model described above. In a first approximation, the classifier’s overall performance is obtained for each pair of resampling techniques, choosing the best-performing classification algorithm for each category (Approach 1). Then, an improvement in the combination of the models is proposed by selecting the resampling technique that produces the best results for each category, considering this approach as the final proposal of this study (Approach 2). Before showing the results, it is necessary to note the following discussion regarding the composition of the binary models generated for each category.

Table 4 shows the resampling operations applied depending on the binary model generated. The rows indicate the binary model to be generated, and the columns indicate the resampling operation performed for each category that makes up the final one-vs-rest dataset. The decision to oversample or undersample depends on the number of samples generated, following the methodology detailed in Algorithm 1. For example, to create the Analysis model, the Worms category should be oversampled since there are fewer than the required number of samples (222) in the training set (130). The rest will be undersampled because they exceed the calculated sample threshold.

Table 4 Comparison of resampling techniques for each category in terms of macro-F1 using NB.

	Composition		
Models	An.	Ba.	DoS	Ex.	Fu.	Ge.	No.	Re.	Sh.	Wo.	
Analysis	–	US	US	US	US	US	US	US	US	OS	
Backdoor	US	–	US	US	US	US	US	US	US	OS	
DoS	US	US	–	US	US	US	US	US	OS	OS	
Exploits	OS	OS	US	–	US	US	US	US	OS	OS	
Fuzzers	OS	OS	US	US	–	US	US	US	OS	OS	
Generic	OS	OS	US	US	US	–	US	US	OS	OS	
Normal	OS	OS	US	US	US	US	–	US	OS	OS	
Reconnaissance	US	US	US	US	US	US	US	–	OS	OS	
Shellcode	US	US	US	US	US	US	US	US	–	US	
Worms	US	US	US	US	US	US	US	US	US	–	

For the two-stage model construction, it is observed that the application of undersampling is more frequent than oversampling operations. Consequently, the generation of datasets will involve the selection of more representative samples for each class rather than generating a large number of artificial samples. For this reason, the choice of undersampling techniques will have a more significant impact on the final performance of the model. Reviewing Table 4 in depth, it is found that the DoS, Exploits, Fuzzers, Generic, Normal and Reconnaissance categories will not be oversampled in any case. Precisely, these are the categories with the highest number of samples. On the other hand, the Analysis, Backdoor, Shellcode, and Worms classes will be oversampled in most experiments due to the minor traffic collected during these cyberattacks.

Binary models results

Table 5 shows macro-F1 after generating balanced binary models with the different groups of resampling techniques using NB classification algorithm. To evaluate each model, the original test set has been binarized into two classes: one for the category the model seeks to detect and another class for the rest of the traffic. When applying the classical NB classification algorithm, only the Worms class yields better results when no resampling technique is applied. This may be due to the fact that the set of samples labelled as Worms is small and, therefore, the generation of synthetic samples is not a clear improvement. Detecting individual attacks is not very good, producing macro-F1 values around 0.5 for Analysis, Backdoor, Reconnaissance, Shellcode and Worms. Slightly better results are achieved for DoS and Exploits, which reach 0.6128 and 0.6640 for macro-F1, respectively. Regarding legitimate traffic detection, macro-F1 reaches 0.7692 after resampling the training set with Borderline1 and RENN for oversampling and undersampling, respectively.

Table 5 Comparison of resampling techniques for each category in terms of macro-F1 using NB.

	Analysis	Backdoor	DoS	Exploits	Fuzzers	Generic	Normal	Reconnaissance	Shellcode	Worms	
No resampling	0.5237	0.5250	0.5749	0.5703	0.5714	0.8424	0.7427	0.4662	0.4724	0.5027	
SMOTE+RU	0.5236	0.5281	0.5284	0.6361	0.4680	0.8422	0.7250	0.4666	0.4723	0.3998	
SMOTE+CC	0.3252	0.3269	0.6111	0.5789	0.4670	0.8422	0.7250	0.4666	0.4725	0.3930	
SMOTE+NM3	0.3224	0.3336	0.4105	0.6248	0.4322	0.8832	0.6187	0.3654	0.4700	0.4347	
SMOTE+RENN	0.5227	0.4712	0.6038	0.5263	0.4678	0.8406	0.7675	0.4626	0.4722	0.4749	
SMOTE+TKL	0.5213	0.5254	0.5337	0.5724	0.5708	0.8345	0.7395	0.5194	0.4724	0.5022	
Bline1+RU	0.5242	0.5272	0.4925	0.6135	0.4673	0.8416	0.7232	0.4660	0.4723	0.3998	
Bline1+CC	0.3194	0.3170	0.3669	0.6145	0.4598	0.8302	0.7337	0.4560	0.4718	0.3662	
Bline1+NM3	0.3214	0.3327	0.4083	0.6343	0.4299	0.8712	0.6196	0.3591	0.4700	0.4347	
Bline1+RENN	0.5227	0.4712	0.6044	0.5256	0.4676	0.8406	0.7692	0.4620	0.4722	0.4749	
Bline1+TKL	0.5213	0.5254	0.5332	0.5726	0.5706	0.8345	0.7425	0.5177	0.4724	0.5022	
ADASYN+RU	0.5235	0.5280	0.6128	0.6640	0.5423	0.8429	0.7284	0.3775	0.4723	0.3998	
ADASYN+CC	0.3195	0.3172	0.6118	0.6073	0.5263	0.8312	0.7331	0.4919	0.4718	0.3662	
ADASYN+NM3	0.3223	0.3335	0.3326	0.6014	0.4728	0.8907	0.7048	0.3509	0.4700	0.4347	
ADASYN+RENN	0.5227	0.4712	0.6077	0.5273	0.5245	0.8409	0.7332	0.4897	0.4722	0.4749	
ADASYN+TKL	0.5213	0.5254	0.5847	0.5720	0.5156	0.8349	0.7337	0.4892	0.4724	0.5022	
Note:

Bold text highlights the best pair of resampling techniques for each kind of traffic.

Table 6 considers macro-F1 as an evaluation metric for the binary models obtained from the training of the KNN classification algorithm. As in the previous experiment, this metric aims to evaluate the classification ability of the specialized model between one type of network traffic and the rest. Analogous to the previous models, Worms obtains macro-F1 when the imbalance is not solved. The same applies in the case of Exploits, Reconnaissance and Shellcode classes. However, for these three classes, there are resampling methods that achieve similar performance. When training KNN for traffic detection collected in UNSW-NB15, the models increase the generalization capability for Backdoor, DoS and Fuzzers when using ADASYN+RENN. Regarding attack detection, the Normal model achieves 0.8731 for macro-F1 using SMOTE+TKL.

Table 6 Comparison of resampling techniques for each category in terms of macro-F1 using KNN.

	Analysis	Backdoor	DoS	Exploits	Fuzzers	Generic	Normal	Reconnaissance	Shellcode	Worms	
No resampling	0.5084	0.5104	0.5368	0.7908	0.6257	0.9873	0.8203	0.8001	0.6608	0.6110	
SMOTE+RU	0.5127	0.4917	0.5590	0.7420	0.6168	0.9862	0.8647	0.6904	0.4727	0.4579	
SMOTE+CC	0.5149	0.4871	0.5537	0.7420	0.6213	0.9862	0.8647	0.6706	0.4727	0.4623	
SMOTE+NM3	0.3823	0.2829	0.2803	0.3500	0.4237	0.8488	0.5970	0.2221	0.1968	0.3303	
SMOTE+RENN	0.3823	0.2829	0.2803	0.3500	0.4237	0.8488	0.5970	0.2221	0.1968	0.3303	
SMOTE+TKL	0.5260	0.5196	0.6726	0.7460	0.6191	0.6560	0.8731	0.6162	0.5502	0.5427	
Bline1+RU	0.5126	0.4916	0.5581	0.7421	0.6154	0.9863	0.8655	0.6904	0.4727	0.4579	
Bline1+CC	0.5169	0.4949	0.5772	0.7494	0.6232	0.9871	0.8697	0.6877	0.4753	0.4629	
Bline1+NM3	0.3823	0.2833	0.2785	0.3449	0.4214	0.8440	0.5949	0.2238	0.1968	0.3303	
Bline1+RENN	0.5260	0.5196	0.6720	0.7446	0.6182	0.6561	0.8730	0.6160	0.5502	0.5427	
Bline1+TKL	0.5071	0.5104	0.5401	0.7737	0.6237	0.9875	0.8268	0.7983	0.6534	0.6090	
ADASYN+RU	0.5137	0.4920	0.5629	0.7416	0.6236	0.9868	0.8646	0.6825	0.4727	0.4579	
ADASYN+CC	0.5169	0.4950	0.5808	0.7493	0.6250	0.9871	0.8685	0.6663	0.4753	0.4629	
ADASYN+NM3	0.3820	0.2825	0.3142	0.3512	0.4321	0.8500	0.5974	0.3113	0.1968	0.3303	
ADASYN+RENN	0.5260	0.5196	0.6794	0.7479	0.6264	0.6561	0.8727	0.6231	0.5502	0.5433	
ADASYN+TKL	0.4964	0.5013	0.5386	0.7731	0.6243	0.9873	0.8259	0.7803	0.6534	0.6090	
Note:

Bold text highlights the best pair of resampling techniques for each kind of traffic.

Table 7 evaluates the detection capability between a class and the rest, considering macro-F1 as the evaluation metric. It shows the results after training models with the SVM algorithm. The optimal hyperplane search obtains the best evaluation using a raw dataset for generic attack traffic. However, it should be noted that performance metrics for this class are very high, and the differences between the different resampling techniques are minor. Furthermore, an interesting finding is that the best-performing resampling method for SVM models is RENN. for majority sample reduction. Regarding suspicious network traffic detection, Borderline1 + RENN yields the best macro-F1 with 0.8528.

Table 7 Comparison of resampling techniques for each category in terms of macro-F1 using SVM.

	Analysis	Backdoor	DoS	Exploits	Fuzzers	Generic	Normal	Reconnaissance	Shellcode	Worms	
No resampling	0.5084	0.5104	0.5368	0.7908	0.6257	0.9865	0.8203	0.4892	0.6608	0.6110	
SMOTE+RU	0.4287	0.4211	0.5069	0.7574	0.5833	0.9825	0.7889	0.5299	0.4032	0.4592	
SMOTE+CC	0.4464	0.4446	0.5411	0.7574	0.5964	0.9852	0.7621	0.5915	0.4533	0.4627	
SMOTE+NM3	0.2463	0.3221	0.2509	0.1805	0.3107	0.8765	0.7097	0.3305	0.1738	0.3949	
SMOTE+RENN	0.5572	0.5438	0.6490	0.7582	0.6030	0.9858	0.8224	0.5905	0.4988	0.4999	
SMOTE+TKL	0.4979	0.4982	0.4897	0.7501	0.5639	0.9861	0.7899	0.4892	0.4988	0.4999	
Bline1+RU	0.4253	0.4210	0.5409	0.7429	0.5839	0.9825	0.7891	0.5250	0.4032	0.4592	
Bline1+CC	0.4118	0.4162	0.5048	0.6521	0.5676	0.9786	0.7950	0.4944	0.3790	0.4597	
Bline1+NM3	0.2351	0.3583	0.2372	0.1407	0.3885	0.8714	0.6457	0.3731	0.2086	0.3615	
Bline1+RENN	0.5546	0.5435	0.6490	0.7667	0.6044	0.9859	0.8528	0.6226	0.4988	0.4999	
Bline1+TKL	0.4979	0.4982	0.4897	0.7499	0.5715	0.9861	0.7898	0.4892	0.4988	0.4999	
ADASYN+RU	0.4312	0.4211	0.6456	0.7578	0.5798	0.9852	0.8016	0.4892	0.4533	0.4627	
ADASYN+CC	0.4136	0.4158	0.6282	0.6808	0.5534	0.9809	0.7669	0.4892	0.3790	0.4597	
ADASYN+NM3	0.2455	0.3204	0.3132	0.2921	0.5085	0.8766	0.6983	0.4925	0.1738	0.3949	
ADASYN+RENN	0.5542	0.5435	0.6500	0.7656	0.6043	0.9825	0.7853	0.6012	0.5188	0.5045	
ADASYN+TKL	0.4979	0.4982	0.4897	0.7501	0.5057	0.9862	0.7898	0.4892	0.4988	0.4999	
Note:

Bold text highlights the best pair of resampling techniques for each kind of traffic.

Table 8 groups the models built after training DT. Following the same criterion, macro-F1 is considered for estimating the most accurate model. Again, Fuzzers, Generic and Worms are more easily distinguishable from the other samples when the imbalance problem is not addressed. Nevertheless, it should be noted that most classes achieve the best performance when the DT algorithm is considered. ADASYN + TKL are the best combination of oversampling and undersampling techniques for Exploits, Reconnaissance and Shellcode, achieving macro-F1 of 80.14, 92.07 and 73.24, respectively. The Normal model, responsible for distinguishing between legitimate and illegitimate traffic, achieves 91.14 for macro-F1 when ADASYN+CC is performed.

Table 8 Comparison of resampling techniques for each category in terms of macro-F1 using DT.

	Analysis	Backdoor	DoS	Exploits	Fuzzers	Generic	Normal	Reconnaissance	Shellcode	Worms	
No resampling	0.4976	0.5320	0.5621	0.8000	0.6464	0.9879	0.8581	0.8482	0.7194	0.7499	
SMOTE+RU	0.5134	0.5025	0.6189	0.7898	0.6226	0.9817	0.8833	0.8631	0.5431	0.4885	
SMOTE+CC	0.5089	0.5010	0.6149	0.7635	0.6233	0.9251	0.8901	0.8585	0.5488	0.4807	
SMOTE+NM3	0.3667	0.2962	0.2335	0.5276	0.4131	0.7162	0.6258	0.4267	0.4911	0.4476	
SMOTE+RENN	0.5248	0.5212	0.6742	0.7697	0.6217	0.9796	0.9011	0.7027	0.6137	0.7253	
SMOTE+TKL	0.4962	0.5244	0.5536	0.7955	0.6353	0.9858	0.8667	0.9004	0.7324	0.7141	
Bline1+RU	0.4378	0.5011	0.6221	0.7826	0.6095	0.9834	0.8941	0.8695	0.5431	0.4885	
Bline1+CC	0.5091	0.4414	0.5549	0.7561	0.6073	0.9825	0.8663	0.7605	0.4868	0.4518	
Bline1+NM3	0.3667	0.2976	0.3813	0.4017	0.4152	0.7289	0.6333	0.4410	0.4911	0.4476	
Bline1+RENN	0.5262	0.5219	0.6707	0.7756	0.6185	0.9784	0.8978	0.7031	0.6137	0.7253	
Bline1+TKL	0.4958	0.5244	0.5627	0.7915	0.6125	0.9587	0.8720	0.9093	0.7324	0.7141	
ADASYN+RU	0.5126	0.4939	0.6353	0.7882	0.5863	0.9835	0.8946	0.8789	0.5431	0.4885	
ADASYN+CC	0.4798	0.4363	0.5662	0.7591	0.6165	0.9832	0.9114	0.7912	0.4868	0.4518	
ADASYN+NM3	0.3525	0.3410	0.3322	0.2909	0.3771	0.7154	0.8596	0.3845	0.4911	0.4476	
ADASYN+RENN	0.5259	0.5215	0.6794	0.7879	0.6194	0.9755	0.8960	0.7096	0.6137	0.7253	
ADASYN+TKL	0.4958	0.5244	0.5631	0.8014	0.6235	0.9487	0.8667	0.9207	0.7324	0.7141	
Note:

Bold text highlights the best pair of resampling techniques for each kind of traffic.

Approach 1: evaluation over resampling methods

Following the model generation explained in “Generation of Classification Models”, Fig. 8 first approach displays how binary model selection is performed when oversampling and undersampling techniques are used to generate a resampled dataset. Table 9 collects the evaluation metrics calculated for the 16 experiments carried out. These 16 experiments are the result of combining the three oversampling methods (SMOTE, Borderline-1 SMOTE and ADASYN) with the five undersampling methods (random undersampling, cluster centroids, Near-Miss 3, repeated edited nearest neighbors and Tomek Links). Evaluation in imbalanced domains has some peculiarities. Metrics such as accuracy can lead to errors in the interpretation of results. Precision, recall, F1 and G-mean metrics have been collected to solve this problem. To test each of the 16 experiments, the test set described in Moustafa & Slay (2015) is used.

Figure 8 Selection of binary models for evaluating a single pair of resampling techniques.

Highlighted cubes represent best performing binary models in terms of macro-F1.

Table 9 Comparison of resampling techniques using the two-stage proposal.

	Accuracy	Precision	Recall	F1	G-mean	
No resampling (Baseline)	0.7207	0.8071	0.7207	0.7537	0.8242	
SMOTE+RU	0.7333	0.8078	0.7333	0.7627	0.8243	
SMOTE+CC	0.7107	0.8131	0.7107	0.7461	0.8009	
SMOTE+NM3	0.4412	0.6200	0.4412	0.3803	0.4590	
SMOTE+RENN	0.7719	0.7918	0.7719	0.7769	0.8428	
SMOTE+TKL	0.7379	0.8101	0.7379	0.7660	0.8321	
Bline1+RU	0.7450	0.8054	0.7450	0.7690	0.8280	
Bline1+CC	0.6941	0.7739	0.6941	0.7169	0.7749	
Bline1+NM3	0.4328	0.3976	0.4328	0.3538	0.4028	
Bline1+RENN	0.7696	0.7887	0.7696	0.7738	0.8390	
Bline1+TKL	0.7315	0.8017	0.7315	0.7599	0.8253	
ADASYN+RU	0.7392	0.8166	0.7392	0.7702	0.8280	
ADASYN+CC	0.7238	0.8130	0.7238	0.7511	0.8053	
ADASYN+NM3	0.4931	0.6698	0.4931	0.4686	0.5492	
ADASYN+RENN	0.7719	0.7861	0.7719	0.7746	0.8389	
ADASYN+TKL	0.7368	0.8083	0.7368	0.7648	0.8298	

Unsurprisingly, undersampling techniques lead to a more significant impact on the classifier’s performance than oversampling algorithms. As shown in Table 4, undersampling is carried out more frequently. Our experiments demonstrate that each undersampling technique produces similar evaluation metrics among the three oversampling algorithms. In this experiment, oversampling techniques have a slight impact on classification tasks.

When oversampling is applied, either SMOTE, Borderline-1 or ADASYN, the best performing undersampling method is RENN, obtaining an F1 around 0.77 and a G-mean around 0.84. When using RENN with the oversampling methods, SMOTE + RENN shows the best results with 0.7708, 0.7918, 0.7708, 0.7760, 0.8428 for accuracy, precision, recall, F1 and G-mean, respectively. The rationale behind RENN’s performance is that it works by removing traffic samples whose category differs from a majority subset. Removal of these potential outliers outperforms the rest of the methods.

G-mean indicates true negative rate (specificity) and true positive rate (sensitivity). Although many research works focus on evaluating their methods using accuracy, G-mean allows considering class imbalance when measuring the performance of a classifier. Similarly to F1, G-mean achieves higher values when using RENN with any of the studied oversampling algorithms. In these experiments, this metric ranges from 0.8389 to 0.8428, indicating a good balance between individual accuracies among ten traffic categories. Regarding NM3 experiments, G-mean reaches 0.4686, which is the best case for this undersampling method when combined with ADASYN. These metrics evidence severe misclassification in some classes.

Figure 9 graphically compares the 15 experiments performed and, in turn, how their performance ranks against not applying any balancing technique. Note that this baseline performance has been obtained through the same experimental design but without using resampled datasets for each traffic category. Baseline model achieves 0.7207, 0.8071, 0.7207, 0.7537 and 0.8242 for accuracy, precision, recall, F1-score and G-mean, respectively. It can be seen that nine of the 15 experiments outperform the system built from unbalanced datasets (baseline performance). These nine experiments correspond to SMOTE, Borderline1 and ADASYN techniques with RU, RENN and TKL. Therefore, these three undersampling methods improve the base performance in combination with any of the three oversampling methods.

Figure 9 Comparison of resampling techniques against baseline approach in terms of F1-score for multi-class classification.

The improvement achieved by deleting Tomek links from the majority classes demonstrates that UNSW-NB15 contains noisy samples for several traffic categories. As discussed, this method reduces noise by removing adjacent samples from different classes. Contrary to expectations, this study demonstrates that the application of RU supposes a significant improvement in terms of F1 compared to baseline performance. Another interesting finding is that NM3 undersampling is not recommended when working with imbalanced problems in the network traffic domain. Low results may be explained by the fact that since these techniques remove relevant traffic data during the undersampling process, classifiers cannot establish well-defined borders between different cyberattack samples.

Approach 2: proposed classification model

A second approach is proposed to improve the performance of the previous experiments. It consists of the combination of the best models for each traffic category collected in UNSW-NB15. Unlike the experiments discussed above, where an oversampling and an undersampling technique were used to generate all the specialized models and create the two-step model, this second experiment selects the best-performing model. It compares their performance for each pair of resampling operations. This improvement supposed this approach to be considered as the proposal of this study. Since this is a two-phase model, the first step will be to evaluate the performance of the binary models in charge of attack detection (Normal class), i.e., the one able to distinguish between legitimate and illegitimate traffic. For this purpose, based on the macro F1-score evaluation metric, the best model is chosen. This metric, which relates precision and recall, aims to minimize the number of samples incorrectly classified as legitimate traffic so that the remaining nine models can reevaluate these at the second classification level. Figure 10 contains the evaluation for the first step of the model. In particular, generating the binary model for legitimate traffic detection from ADASYN + CC results in the system with the best global classification capacity, reaching a value slightly above 0.91.

Figure 10 Comparison of resampling techniques against baseline approach in terms of macro F1-score for first step of the two-stage model.

Once the model has been selected for the first phase, the second classification level is composed of the sampling models that achieve the best performance individually for each category. Thus, the best-performing algorithm and sampling techniques are selected for each type of traffic. Since the construction of these models aims to distinguish between one class and the rest, the evaluation methodology used has assigned the same weight to both classes (macro-F1), i.e., the evaluation is not affected by the number of samples of each class.

For each pair of resampling methods, Table 10 shows macro-F1 for the best algorithm (among DT, NB, KNN and SVM) for each kind of attack present in UNSW-NB15. In most cases, resampling yields slightly better results than original data without balancing. Unlike previous experiments, these results showed that RENN is not the best choice for undersampling imbalanced data. Depending on data distribution, it is necessary to find the pair of resampling methods that fit better for training a model able to distinguish one traffic from the rest. Generic, Normal and Reconnaissance are the most easily identifiable categories, obtaining 0.9879, 0.9114 and 0.9207 for macro-F1, respectively. Generic traffic achieves the best result without resampling. An explanation for this might be that there are enough samples in the train set and no fuzzy borders between this class and the rest. Regarding Normal and Generic traffic, generating a balanced train set using ADASYN produces better results than raw one-vs-rest train sets. Analysis and Backdoor OvR specialized models do not produce good results, but a performance increase is found when SMOTE + RENN is used in comparison with no resampling approach.

Table 10 Comparison of resampling techniques for each category in terms of macro-F1 using best algoritms for each traffic.

	Analysis	Backdoor	DoS	Exploits	Fuzzers	Generic	Normal	Reconnaissance	Shellcode	Worms	
No resampling	0.5237	0.5320	0.5749	0.8000	0.6464	0.9879	0.8581	0.8482	0.7194	0.7499	
SMOTE+RU	0.5236	0.5281	0.6189	0.7898	0.6226	0.9862	0.8833	0.8631	0.5431	0.4885	
SMOTE+CC	0.5149	0.5010	0.6111	0.7635	0.6233	0.9862	0.8901	0.8585	0.5488	0.4807	
SMOTE+NM3	0.3823	0.3336	0.4105	0.6248	0.4322	0.8832	0.7097	0.4267	0.4911	0.4476	
SMOTE+RENN	0.5572	0.5438	0.6742	0.7697	0.6217	0.9858	0.9011	0.7027	0.6137	0.7253	
SMOTE+TKL	0.5213	0.5254	0.5536	0.7955	0.6353	0.9873	0.8667	0.9004	0.7324	0.7141	
Bline1+RU	0.5242	0.5272	0.6221	0.7826	0.6154	0.9863	0.8941	0.8695	0.5431	0.4885	
Bline1+CC	0.5169	0.4949	0.5772	0.7561	0.6232	0.9871	0.8697	0.7605	0.4868	0.4629	
Bline1+NM3	0.3823	0.3583	0.4083	0.6343	0.4299	0.8714	0.6457	0.4410	0.4911	0.4476	
Bline1+RENN	0.5546	0.5435	0.6720	0.7756	0.6185	0.9859	0.8978	0.7031	0.6137	0.7253	
Bline1+TKL	0.5213	0.5254	0.5627	0.7915	0.6237	0.9875	0.8720	0.9093	0.7324	0.7141	
ADASYN+RU	0.5235	0.5280	0.6456	0.7882	0.6236	0.9868	0.8946	0.8789	0.5431	0.4885	
ADASYN+CC	0.5169	0.4950	0.6282	0.7591	0.6250	0.9871	0.9114	0.7912	0.4868	0.4629	
ADASYN+NM3	0.3820	0.3410	0.3326	0.6014	0.5085	0.8907	0.8596	0.4925	0.4911	0.4476	
ADASYN+RENN	0.5542	0.5435	0.6794	0.7879	0.6264	0.9755	0.8960	0.7096	0.6137	0.7253	
ADASYN+TKL	0.5213	0.5254	0.5847	0.8014	0.6243	0.9873	0.8667	0.9207	0.7324	0.7141	
Note:

Bold text highlights the best pair of resampling techniques for each kind of traffic.

Table 11 lists detailed information on the models that make up the final system. The resampling techniques used for sample generation in constructing each specialized binary model and the best-performing algorithm are indicated. For the first classification phase, the Normal model will be in charge of detecting legitimate behavior. For this purpose, ADASYN + CC is applied, which yields the best classification results along with the decision trees algorithm. The best-performing algorithms at the second classification level were SVM and decision trees. The tuned hyperparameters used for generating each model are indicated to facilitate the replication of the experiments. Notice that the environment should be configured Python 3.8 and Scikit-Learn.

Table 11 Binary models configurations for proposed classification model.

	Resampling technique	Algorithm	Hyperparameters	
Analysis	SMOTE+RENN	SVM	C = 10.0, probability = True, random_state = 42	
Backdoor	SMOTE+RENN	SVM	C = 10.0, probability = True, random_state = 42	
DoS	ADASYN+RENN	Decision trees	max_depth = 22, min_samples_leaf = 2, random_state = 42	
Exploits	ADASYN+TKL	Decision trees	criterion = ‘entropy’, max_depth = 20, max_features = ‘auto’, min_samples_leaf = 4, random_state = 42	
Fuzzers	–	Decision trees	max_depth = 20, max_features = ‘auto’, min_samples_leaf = 3, random_state = 42	
Generic	–	Decision trees	max_depth = 22, max_features = ‘auto’, min_samples_leaf = 4, random_state = 42	
Normal	ADASYN+CC	Decision trees	max_depth = 20, min_samples_split = 4, random_state = 42	
Reconnaissance	ADASYN+TKL	Decision trees	max_depth = 20, min_samples_leaf = 4, random_state = 42	
Shellcode	ADASYN+TKL	Decision trees	criterion = ‘entropy’, max_depth = 21, min_samples_split = 4, random_state = 42	
Worms	–	Decision trees	max_depth = 20, min_samples_leaf = 4, random_state = 42	

A statistical procedure has been carried out to determine if the performance of the second approach can be considered statistically significant. As a reminder, the baseline consists of a hybrid two-stage ensemble without dealing with class imbalance. In addition, four new non-hybrid two-stage ensembles are built employing classification algorithms widely used in the literature (NB, KNN, SVM, DT). The main difference with the baseline approach is that each of these four classifiers is built using only one algorithm per traffic category. Thus, the NB-based model consists of 10 binary models obtained through the training of the Naïve Bayes algorithm. The same occurs for the three remaining models. This process aims to verify if the obtained results are statistically significant compared to six more classifiers. A 10-fold cross validation is executed for each classifier to estimate the average F1.

The statistical analysis was conducted for seven experiments with 10 paired samples as shown in Table 12. The tests’ family-wise significance level (p-value) is set to 0.05. The first step is to conduct the Shapiro & Wilk (1965) test to check normality. It has failed to reject the null hypothesis that the population is normal for all populations (minimal observed p-value = 0.241). Therefore, it is assumed that all populations are normal. Bartlett’s test is applied (Arsham & Lovric, 2011) for homogeneity, and it failed to reject the null hypothesis (p = 0.761) that the data is homoscedastic. This implies that data is homoscedastic, so that assumption is considered. There are more than two populations and all are normal and homoscedastic so repeated measures ANOVA (Kaufmann & Schering, 2014) are used to determine if there are any significant differences between the mean values of the populations. If the results of the ANOVA test are significant, post-hoc Tukey HSD tests (Tukey, 1949) are needed to infer which differences are significant. Each population’s mean value (M) and standard deviation (SD) are reported. Populations are significantly different if their confidence intervals are not overlapping. The null hypothesis is rejected ( p=1∗10e−7) for repeated measures ANOVA that there is a difference between the mean values of the populations NB (M = 0.595 + −0.013, SD = 0.022), SVM (M = 0.710 + −0.013, SD = 0.016), KNN (M = 0.724 + −0.013, SD = 0.023), DT (M = 0.746 + −0.013, SD = 0.023), Baseline (M = 0.749 + −0.013, SD = 0.018), Approach 1 (M = 0.780 + −0.013, SD = 0.014), and proposed model (M = 0.797 + −0.013, SD = 0.018). Therefore, it is assumed that there is a statistically significant difference between the mean values of the populations. Based on the post-hoc Tukey HSD test, the following groups are considered to have no significant differences: DT and Baseline. These relationships may be explained by the fact that the Baseline two-step ensemble classifier contains several DT binary models also included in the DT-Based approach. All other differences are significant. This implies that present results are statistically significant to be compared with state-of-art approaches.

Table 12 Statistical analysis over UNSW-NB15.

	Meanrank	Mean	Std	CI_lower	CI_upper	Effect_size	
Proposal	1.2	0.7968	0.0176	0.7837	0.8100	10.2334	
Approach 1	2.0	0.7799	0.0141	0.7668	0.7931	10.1244	
Baseline	3.5	0.7486	0.0180	0.7355	0.7617	7.7197	
DT	3.9	0.7463	0.0229	0.7332	0.7595	6.7989	
KNN	4.9	0.7237	0.0226	0.7106	0.7369	5.8177	
SVM	5.5	0.7096	0.0162	0.6965	0.7228	5.9929	
NB	7.0	0.5947	0.0216	0.5815	0.6078	0	

Once the proposal demonstrates to be statistically significant, Table 13 contains the confusion matrix obtained from the two-phase model built from binary classifiers shown in Table 11 using the test set. Attending the matrix’s main diagonal, the model performs well for the categories Exploits, Generic, Normal, Reconnaissance and Shellcode, with many correctly classified samples. In an intrusion detection system, attack detection is critical. In this regard, it is important to emphasize the number of samples correctly classified as Normal, which is 33,005 out of 37,000. Regarding the second level classification, in charge of categorising cyberattacks, the matrix shows how much of the traffic generated from other attacks is classified as DoS. Two reasons could explain this behavior. On the one hand, the environment where this traffic has been generated has suffered few changes during the attacks. Therefore, the samples used to generate the DoS lack helpful information to distinguish these attacks from the rest. On the other hand, the combination of ADASYN + RENN may not be optimal. While ADASYN tries to generate new synthetic samples that are hard to learn (near borderline) for minority attacks, RENN removes the majority of samples targeted with a label different from the surrounding ones. If tested samples belonging to undersampled classes are located near the borderline, the classification task is more prone to failure. Regarding traffic classification labelled as Analysis or Backdoor, the task is challenging due to data. These findings are also reported in other studies (Khan et al., 2019; Zhang et al., 2020; Manimurugan, 2021).

Table 13 Confusion matrix for proposed classification model.

	Analysis	Backdoors	DoS	Exploits	Fuzzers	Generic	Normal	Reconnaissance	Shellcode	Worms	
Analysis	21	3	556	55	26	0	16	0	0	0	
Backdoors	18	7	462	58	30	1	3	0	4	0	
DoS	68	37	2,832	745	227	10	111	21	36	2	
Exploits	118	114	3,337	6,416	317	35	466	217	106	6	
Fuzzers	109	76	1,231	470	1,541	7	2,483	16	127	2	
Generic	32	16	180	298	36	18,230	55	0	21	3	
Normal	591	78	401	744	2,010	28	33,005	19	123	1	
Reconnaissance	19	4	297	302	33	1	61	2,745	32	2	
Shellcode	14	1	5	37	29	1	16	3	272	0	
Worms	2	0	1	13	0	1	1	1	0	25	

From the confusion matrix shown in Table 13, the proposed model’s performance can be estimated by calculating the aforementioned evaluation metrics.

Table 14 shows the comparison between (1) a non-hybrid two-stage ensemble employing four classification algorithms widely used in the literature (NB, KNN, SVM, DT) without resampling, (2) a hybrid two-stage ensemble without dealing with class imbalance (Baseline) and (3) Proposal. These three sets of results are compared against some existing state-of-the-art works. Since binary classification corresponds to the first classification step, only one model for benign traffic needs to be tested. For this reason, the DT-based model is chosen as the baseline for binary classification.

Table 14 Comparison between baseline and proposed classification model.

Approach	Classification	Accuracy	Precision	Recall	F1	G-mean	
Khan et al. (2019)	Binary	0.8971	0.8670	0.9246	0.8697	–	
Zhou et al. (2021)	Binary	–	0.8600	0.9780	0.9070	–	
Liu, Gao & Hu (2021)	Binary	0.8589	–	–	–	–	
NB	Binary	0.7520	0.7579	0.7403	0.7427	0.7313	
KNN	Binary	0.8312	0.8678	0.8146	0.8203	0.7980	
SVM	Binary	0.8075	0.8676	0.7862	0.7898	0.7570	
DT (baseline)	Binary	0.8637	0.8828	0.8522	0.8581	0.8446	
Proposal	Binary	0.9125	0.9123	0.9106	0.9114	0.9104	
Kasongo & Sun (2020a)	Multi-class	0.7716	–	–	–	–	
Kasongo & Sun (2020b)	Multi-class	0.7751	–	–	–	–	
Al-Turaiki & Altwaijry (2021)	Multi-class	0.6946	0.8400	0.6900	0.7400	–	
NB	Multi-class	0.6071	0.5976	0.6071	0.5813	0.6556	
KNN	Multi-class	0.7343	0.7725	0.7343	0.7475	0.8100	
SVM	Multi-class	0.6934	0.7563	0.6934	0.7141	0.7979	
DT	Multi-class	0.7245	0.7998	0.7245	0.7526	0.8257	
Baseline	Multi-class	0.7207	0.8071	0.7207	0.7537	0.8242	
Proposal	Multi-class	0.7906	0.8154	0.7906	0.7972	0.8550	
Note:

Bold text highlights the best evaluation metrics across related works and proposed work. The first set of rows is for binary classification and bold values are used for best accuracy, precision, recall, F1 and G-mean across binary classification. The second set of rows is for multiclass classification and bold values are used for best accuracy, precision, recall, F1 and G-mean across multiclass classification.

In the current study, when the model is made up of unbalanced models, the two-step ensemble does not produce the best results compared to other proposals in the literature. However, after selecting the correct balanced models and following the two-stage schema, an improvement is observed in both steps. For binary classification, taking baseline as a reference, a significant improvement is observed in the first stage, around 5% for F1-score and 7% for G-mean, which aims to distinguish between attack and legitimate behaviour. In the second stage, where suspicious traffic is classified among the nine remaining categories, results lead to similar conclusions improving over 4% for F1-score and 3% for G-mean compared to baseline. Compared with other state-of-art works, the proposed contribution delivers significantly better results in terms of F1. These results prove that choosing the suitable resampling operations and classification algorithm for a binary OvR dataset can improve performance when developing a two-step multi-class classification.

Conclusions

This article presents a two-stage AI model for detecting and classifying cyberattacks in industrial environments leveraging generic network traffic data to cover a broad spectrum of common cyber threats. To accomplish this, a comprehensive study of several resampling algorithms for a highly imbalanced set of cyberattacks is conducted. To address this problem, where most data belongs to legitimate traffic, eight resampling algorithms are evaluated: three for oversampling (SMOTE, Borderline-1, and ADASYN) and five for undersampling (random undersampling, cluster centroids, Near-Miss3, repeated edited nearest neighbors and Tomek Links). These resampling techniques are evaluated and included in developing a multi-class classifier system of network traffic. A total of 640 models have been developed and evaluated using one classification algorithm and a pair of resampling techniques (one oversampling and one undersampling). The proposal includes the best combination of oversampling and undersampling methods. The findings underscore the utility of integrating resampling algorithms to enhance classification performance significantly, a strategy that holds considerable promise for improving cybersecurity measures in industrial settings.

The implications of our research are far-reaching for implementing AI-driven defense mechanisms against cyber threats in real-world systems. The performance of the proposed model can significantly bolster the security posture of such environments, enabling more precise detection and categorization of cyberattacks. The real-world applicability of our model extends across several vital areas within computer security, offering substantial improvements in intrusion detection systems, malware classification, phishing detection, anomaly detection in network traffic, spam and botnet detection, and insider threat detection. By accurately classifying diverse cyber threats, our model empowers security systems to deploy effective countermeasures, enhancing the defense of industrial networks against the evolving landscape of cyber threats. However, deploying our proposed system requires consideration of the time required for model training and the sample modification processes employed by the selected resampling algorithms.

In future work, efficiency will be addressed from two perspectives: first, study the impact of cost-sensitive learning for binary classifiers against the use of resampling algorithms. Second, in-depth research will be conducted to evaluate the effect of different feature selection methods. This focus on enhancing both the accuracy and efficiency of cyberattack detection models promises to contribute significantly to developing more robust cybersecurity infrastructures in real-world scenarios.

Code availability

All code was implemented in Python using Scikit-Learn, Numpy and Pandas. The source code for binary models generation and the two-step ensemble model are available at GitHub (https://github.com/UniversidadExtremadura/resampling-algorithms-for-cyberattack-classification). Preprocessed dataset used in our experiments has been published in Zenodo and it is available at https://doi.org/10.5281/zenodo.10714999.

Additional Information and Declarations

Competing Interests

Author Contributions

Data Availability

The authors declare that they have no competing interests.

Óscar Mogollón Gutiérrez conceived and designed the experiments, performed the experiments, analyzed the data, performed the computation work, prepared figures and/or tables, authored or reviewed drafts of the article, and approved the final draft.

José Carlos Sancho Núñez conceived and designed the experiments, performed the experiments, analyzed the data, performed the computation work, prepared figures and/or tables, authored or reviewed drafts of the article, and approved the final draft.

Mar Ávila conceived and designed the experiments, performed the experiments, analyzed the data, performed the computation work, prepared figures and/or tables, authored or reviewed drafts of the article, and approved the final draft.

Andrés Caro conceived and designed the experiments, performed the experiments, analyzed the data, performed the computation work, prepared figures and/or tables, authored or reviewed drafts of the article, and approved the final draft.

The following information was supplied regarding data availability:

The source code for binary models generation and the two-step ensemble model are available at GitHub and Zenodo:

- https://github.com/UniversidadExtremadura/resampling-algorithms-for-cyberattack-classification.

- Mogollón-Gutiérrez, Ó. (2024). resampling-algorithms-for-cyberattack-classification. Zenodo. https://doi.org/10.5281/zenodo.10820235.

The preprocessed dataset is available at Zenodo: Mogollón-Gutiérrez, Ó. (2024). UNSW-NB15 (no header) [Data set]. Zenodo. https://doi.org/10.5281/zenodo.10714999.

The original UNSW-NB15 Dataset is available at: https://research.unsw.edu.au/projects/unsw-nb15-dataset.

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
