# Peer review of "A detailed study of resampling algorithms for cyberattack classification in engineering applications"

_PeerJ Computer Science, doi:10.7717/peerj-cs.1975_

## Round 0.1 · original submission · Minor Revisions

Dear authors,

Your manuscript requires Minor Revisions.

**Language Note:** PeerJ staff have identified that the English language needs to be improved. When you prepare your next revision, please either (i) have a colleague who is proficient in English and familiar with the subject matter review your manuscript, or (ii) contact a professional editing service to review your manuscript. PeerJ can provide language editing services - you can contact us at copyediting@peerj.com for pricing (be sure to provide your manuscript number and title). – PeerJ Staff

Reviewer 1 ·

Basic reporting

This paper introduces a novel system designed to address class imbalance problems. The authors conducted experiments testing eight resampling methods, employing one-vs-rest binary models with various machine learning algorithms, and developing two-stage classification systems. The evaluation of these methods was performed through a case study in the cybersecurity domain. The utilized dataset comprises ten categories, with one representing normal network traffic and the remaining nine representing distinct types of cyberattacks. The inherent imbalance of the dataset makes it particularly compelling for this study.
In the concluding section of the study, the authors offer a comprehensive explanation of the results. Following the execution of 640 experiments, they concluded that resampling algorithms prove highly effective in addressing cyberattack classification tasks. Furthermore, they inferred that the amalgamation of multiple resampling algorithms significantly enhances the performance of the models.
I commend the authors for the clarity in describing the results, the quality of the figures, and the overall well-structured content. The findings obtained are noteworthy and are likely to have a substantial impact on the scientific community, particularly within the cybersecurity field.
Nevertheless, I would like to address some concerns to enhance the quality of the paper and rectify minor errors. In general terms, I recommend a minor revision for this manuscript.

Experimental design

No comment

Validity of the findings

No comment

Additional comments

I've made a few tweaks for clarity and flow:

• In the section discussing previous research, it is recommended to replace instances of "related works" with "related work."

• An interesting approach to addressing imbalanced classes is anomaly detection, particularly in situations where unexpected behaviour corresponds to a harmful event (as observed in this manuscript). One of the most popular techniques for anomaly detection tasks is AutoEncoders. It might be beneficial to include this information in the related work section.

• Around line 140, when referring to the 'ratio' in the UNSW-NB15 dataset, it would be helpful to clarify how the ratio is measured. If there are 10 classes, which one is considered the 'minority,' and why not the others? Providing this clarification is essential.

• In Table 2, 'generic' is categorized as a type of cyberattack. However, due to the similarity between the words 'generic' and 'normal,' there is a potential for confusion. To prevent misunderstanding, it is advisable to highlight the term 'normal.'

• The information presented in Tables 3, 4, 5, 6, and 7 seems somewhat redundant compared to what is provided in Table 1. Consider ways to minimize this redundancy.

• The explanation of various resampling techniques using the synthetically generated dataset is helpful. However, to facilitate a direct comparison with the original data, it would be appropriate to include Figure 1 alongside Figure 2.

• In Table 9, briefly define the terms 'sensitivity' and 'specificity' to enhance reader understanding.

• While I appreciate the availability of code on GitHub, including at least one example with real data and not just raw code would be beneficial for interested readers.

Reviewer 2 ·

Basic reporting

1.The paper is written in clear, professional English that is unambiguous and well-structured, facilitating readability and comprehension.
2. The paper provides a comprehensive background with sufficient references to existing literature, indicating a well-researched context for the study.
3. The structure appears professional and adheres to academic standards with the inclusion of figures, tables, and raw data, as appropriate for the subject matter.
4. The paper appears self-contained, presenting all the relevant results and explanations needed to understand the hypotheses and conclusions.
5. The paper includes clear definitions of terms, detailed proofs, and formal results, which are essential for replicability and academic rigor.

Experimental design

1. The research aligns with the aims and scope of the journal, focusing on cyberattack classification in engineering applications, which is a pressing issue in cybersecurity.
2. The research question is clearly defined, relevant, and meaningful. The paper explicitly states how the research fills an identified knowledge gap by addressing class imbalance in cybersecurity with resampling techniques.
3. The investigation is rigorous, utilizing a large set of models (640 in total) derived from various resampling techniques. However, the manuscript could benefit from a more detailed discussion of the choice of classification algorithms and their appropriateness for the specific types of cyberattacks being studied.
4. The methods are described in sufficient detail to replicate the study, with explicit mention of the software and libraries used (Python 3.8, Pandas, Scikit-Learn).

Validity of the findings

1. The paper does not explicitly assess the impact and novelty of the findings within the broader field. Future work could benefit from a comparison with state-of-the-art models to contextualize the results.
2. The paper encourages meaningful replication, and the rationale behind the study is clear, filling a knowledge gap in cyberattack classification with resampling algorithms.
3. The underlying data have been robustly handled, and the statistical analysis is sound. However, more details on the control measures for data integrity would strengthen the findings.
4. The conclusions are well articulated, linked to the original research question, and supported by the results. Nevertheless, a more in-depth discussion on the potential implications and applications of the findings in real-world scenarios could enhance the conclusion section.

---

## Round 0.2 · accepted · Accept

Dear Dr. Mogollón Gutiérrez,

Thank you for considering the reviewers' comments and making the required changes to your article.

I am happy to inform you that your paper now reaches the required level for publication in PeerJ Computer Science. Thank you for considering our journal for publishing your research papers.

We hope you will continue to consider the journal for publication in your future research.

Sincerely,

M. Emilia Cambronero
Academic Editor of Peer J Computer Science